# Anaplastic Lymphoma Kinase Receptor: Possible Involvement in Anorexia Nervosa

**DOI:** 10.3390/nu15092205

**Published:** 2023-05-06

**Authors:** Simona Dedoni, Maria Scherma, Chiara Camoglio, Carlotta Siddi, Walter Fratta, Paola Fadda

**Affiliations:** 1Section of Neuroscience and Clinical Pharmacology, Department of Biomedical Science, University of Cagliari, 09124 Cagliari, Italy; dedoni@unica.it (S.D.); mscherma@unica.it (M.S.); chiara.camoglio@unica.it (C.C.); carlottasiddi@yahoo.it (C.S.); wfratta@unica.it (W.F.); 2Neuroscience Institute, Section of Cagliari, National Research Council of Italy (CNR), 09042 Cagliari, Italy

**Keywords:** Anorexia Nervosa, activity-based anorexia model, hypothalamus, anaplastic lymphoma kinase receptor

## Abstract

The pathophysiology of Anorexia Nervosa (AN) has not been fully elucidated. Anaplastic lymphoma kinase (ALK) receptor is a protein-tyrosine kinase mainly known as a key oncogenic driver. Recently, a genetic deletion of ALK in mice has been found to increase energy expenditure and confers resistance to obesity in these animals, suggesting its role in the regulation of thinness. Here, we investigated the expression of ALK and the downstream intracellular pathways in female rats subjected to the activity-based anorexia (ABA) model, which reproduces important features of human AN. In the hypothalamic lysates of ABA rats, we found a reduction in ALK receptor expression, a downregulation of Akt phosphorylation, and no change in the extracellular signal-regulated protein kinases 1 and 2 (ERK1/2) phosphorylation. After the recovery from body weight loss, ALK receptor expression returned to the control baseline values, while it was again suppressed during a second cycle of ABA induction. Overall, this evidence suggests a possible involvement of the ALK receptor in the pathophysiology of AN, that may be implicated in its stabilization, resistance, and/or its exacerbation.

## 1. Introduction

Anorexia Nervosa (AN) belongs to the sphere of eating disorders with a lifetime prevalence between 0.3 and 0.6% [1]. AN is characterized by fear of gaining weight, restrictive eating, and consequent body weight loss that impairs physical health associated with high rates of mortality, up to 5.9% [2]. The prolonged exposure to the restrictive diet (self-starvation) in AN patients has a diffuse deleterious impact on the body, inducing bone deterioration, endocrine anomalies, liver dysfunction, anemia, hypoglycemia, and low vitamin D and calcium levels [3,4]. So far, AN etiology is poorly understood. However, several studies have shown the presence of different risk components in AN development, such as genetic predisposition, age, sex, and psychosocial and stress conditions. All these elements can contribute to the onset of the disease. Now, it is well established that AN mainly affects women with a double incidence rate compared to men, and an alarming increase in adolescents in the last decades has been shown [5]. Unfortunately, currently, there are only few effective and targeted therapies, such as the use of antidepressant drugs and psychobiological and nutritional approaches, which lack sufficient specificity [6]. Indeed, almost 40% of individuals receiving treatment for AN end up in hospitals because of ineffectiveness and unfavorable outcomes [7]. In the last two years of the COVID-19 pandemic, the incidence of eating disorders increased, including AN [8,9,10]. To comprehend the potential AN predisposition and to develop measures to lessen vulnerability to this psychiatric disorder, further investigations and research are required. Lately, it has come to light that the regulation of thinness may be influenced by the anaplastic lymphoma kinase (ALK) receptor. The ALK receptor belongs to the tyrosine kinase insulin receptor superfamily (RTK) of 1620 amino acids, and consists of an extracellular ligand-binding domain, a transmembrane domain, and an intracellular tyrosine kinase domain [11]. This receptor was first identified in anaplastic large cell lymphoma (ALCL) as part of the fusion protein NPM-ALK [12], and it is abundantly expressed in different brain areas (cerebellum, hypothalamus, cerebral cortex, olfactory, amygdala, basal ganglia) where it shows a role in the normal development and function of the nervous system [11]. So far, ALK has mainly been studied in correlation with oncogenesis in many types of tumoral pathologies. Specifically, most studies focused on ALK mutations in adult and pediatric malignancies. Along with augmentor-α and -β, other ALK ligands such as pleiotrophin and heparin [13,14,15,16] have been identified, which induce receptor homo-dimerization in different tyrosine residues. The ALK receptor primes distinct downstream pathways: the phosphatidylinositol 3-kinase (PI3K)–Akt, the phospholipase C-γ (PLCγ), the Ras-Extracellular signal-regulated kinase 1/2 (ERK) mitogen-activated protein kinase (MAPK) pathway, and the Janus kinase 3 (JAK3)–STAT3, which contribute to its mitogenic and antiapoptotic activities. Changes to these pathways appear to be the main cause of the protumor response to ALK, and they may interact with one another to contribute to the consequences of ALK activity [17,18,19]. Recently, it has been shown that genetic deletion of ALK resulted in thin mice with marked resistance to diet-induced obesity and increased energy expenditure [20]. Moreover, adiponectin levels were increased in these animals [20]. Orthofer and collaborators also found that ALK expression in hypothalamic neurons controls energy expenditure via sympathetic control of adipose tissue lipolysis. In agreement with this finding, physiological ligands of the ALK receptor such as augmentor α have been shown to regulate body weight in the hypothalamus [21]. Based on the evidence mentioned above, the aim of this study was to investigate a possible role of the ALK receptor in the pathophysiology of AN. The “activity-based anorexia” (ABA) model currently represents a robust and recognized model to study the key symptoms of human AN [22]. Using the ABA model, we evaluated the expression of the ALK protein receptor and the downstream intracellular pathways in the hypothalamus of ABA rats, a key brain region involved in the regulation of food intake whose structure and functionality are altered in AN [23]. The current study attempts to illuminate a novel role for the ALK receptor in AN and offers fresh prospective lines of inquiry for advancing the treatment for this complex illness.

## 2. Experimental Procedure

### 2.1. Animals

Sixty-four naïve Sprague-Dawley female rats (Envigo, Udine, Italy) weighing 125–150 g, PND ~50 at the beginning of the study were used. Rats were housed in a climate-controlled animal room (21 ± 2 °C; 60% humidity; reversed 12 h light/12 h dark cycle, lights on at 12:00 h). Animals were feed with standard rat chow and water ad libitum. We chose young female rats because >90% of patients who suffer AN are adolescent young women [24]. Experimental procedures were carried out according to Italian (D.L. 26/2014) and European Council directives (63/2010) and with the approved animal policies by the Ethical Committee for Animal Experiments at the University of Cagliari (Sardinia, Italy) and the Italian Department of Health (286/2016). We followed all the possible parameters to reduce animal pain and discomfort to the minimum, as well as to decrease the total of experimental animals used. Studies were reported in compliance with the ARRIVE guidelines [25,26].

### 2.2. Apparatus

Standard polycarbonate cages [48 (h) × 32 (w) × 47 (d) cm] or polycarbonate cages equipped with running wheels (35 cm in diameter) were used. Each activity wheel cage was provided with a magnetic switch and LCD revolution counter; the switch continuously counted whole revolutions of the activity wheel (Ugo Basile, Varese, Italy).

### 2.3. Experimental Design

The ABA protocol used in this study was designed as previously described [27,28] (Figure 1). The acclimatization period lasted 1 week, then the animals were divided into 4 groups by randomizing body weight and were individually placed in standard home cages (sedentary rats) or home cages equipped with a running wheel (running rats): (1) ‘control’ rats had 24 h chow access but no access to the activity wheel; (2) ‘restricted’ rats were allowed chow access for 1.5 h/day but had no access to the activity wheel; (3) ‘exercise’ rats had both 24 h chow and activity-wheel access; and (4) ‘ABA’ rats were allowed chow for 1.5 h/day and had 22.5 h activity-wheel access. Rats were adapted to the housing conditions for 7 consecutive days with ad libitum food and running-wheel access (where applicable) (adaptation period). Body weight, food intake, and running-wheel activity (RWA) were recorded every day within 30 min of the start of the 12 h dark cycle to get an established baseline (BL). 

*First ABA induction*. After the adaptation period, ABA and restricted animals were exposed to 6 days of a restricted feeding schedule, which consisted of access to chow for 1.5 h at the onset of the dark phase. Food intake was evaluated by weighing the chow before and after the 1.5 h access period. Running-wheel access by ABA rats was impeded through the 1.5 h feeding period to prevent running on the wheel (a measure of RWA) competing with eating. For the residual time (22.5 h), both ABA and restricted animals had no access to chow, while the ABA rats were allowed access to the wheel. As for the control and exercise groups, they continued to have free access to food, and exercise rats had unlimited access to the wheel. Daily body weight, (control, restricted, exercise, and ABA groups), 24 h food intake (control and exercise groups), and RWA (exercise and ABA groups) were recorded 30 min before the beginning of the 12 h dark period.

*Recovery.* From the last day of the restricted feeding schedule (day 6), ABA and restricted rats underwent a period of weight restoration consisting of 7 days of unrestricted access to chow. Control and exercise animals continued to receive food ad libitum. ABA and exercise rats were allowed unlimited access to the running wheels [22,29]. Body weight (control, restricted, exercise, and ABA groups), food intake (control, restricted, exercise, and ABA groups), and RWA (exercise and ABA groups) were registered every day at least 30 min before the beginning of the 12 h dark period. 

*Second ABA induction.* At the end of the recovery period, ABA and restricted animals were exposed to a second cycle of restricted feeding schedule as in the first cycle [27]. Again, access to food was restricted to 1.5 h at the beginning of the 12 h dark cycle, and chow eating was measured by weighing the chow before and after the period of availability. During the feeding period, the running wheels were locked in the ABA group. Control and exercise rats received 24 h access to chow. Body weight, (control, restricted, exercise, and ABA groups), 24 h chow intake (control and exercise groups), and RWA (exercise and ABA groups) were monitored daily 30 min before the start of the 12 h dark period. 

### 2.4. Plasma Adiponectin Level Assay—ELISA 

Animals were sacrificed at the end of the 12 h light phase on day 6 of the first ABA induction. K_3_EDTA tubes were used to collect trunk blood. It was then centrifuged at 3000× *g* (15 min at 4 ± 2 °C), and then plasma was carefully collected and stored at −20 °C. Adiponectin levels in plasma samples were analyzed using a commercially available ELISA kit according to the manufacturer’s protocols (E-EL-R3012 Rat ADP/Acrp30 ELISA-Kit, Elabscience Biotechnology Inc, Huston, TX, USA). 

### 2.5. Tissue Lysate and Western Blotting

Rats were killed at the end of the 12 h light phase on day 6 of the first ABA induction (*n* = 7 rats per group), on day 7 of the recovery phase (*n* = 7 rats per group), and on day 6 of the second ABA induction (*n* = 7 rats per group). Rat brains were rapidly dissected to isolate the hypothalamus as previously described [28]. The tissues were then stored at −80 °C and processed for solubilization in ice-cold RIPA buffer supplemented with 0.5% and 1% of phosphatase and protease inhibitor cocktail, respectively (Sigma-Merk) (RIPA buffer) [30]. The samples were sonicated for 5 s in an ice bath three times, then centrifuged 10,000× *g* for 10 min. Aliquots of tissue extracts were taken for the Bio-Rad protein assay after samples were transferred to fresh tubes (Bio-Rad Lab, Hercules, CA, USA). Tissue samples, 40 μg for each lysate, were prepared adding the NuPAGE LDS Sample Buffer 4× (Novex ThermoFisher Scientific, Waltham, MA, USA). Next, samples were warmed to 100 °C for 10 min and separated by the polyacrylamide gel (NuPAGE 4–12% Bis-Tris Gel Mini, Novex, ThermoFisher Scientific, Waltham, MA, USA) in the Mini gel tank (ThermoFisher Scientific, Waltham, MA, USA). Internal molecular weight (MW) standards (Prestained Sharpmass vii, Euro Clone Milan, Italy) were run in parallel.

Proteins separated by polyacrylamide gel were transferred to polyvinylidene difluoride membranes (Amersham Biosciences, Piscataway, NJ, USA) by utilizing a semidry device. Membranes were blocked with nonfat dry milk (5%), washed with TBS-Tween 0.05%, and incubated at 4 °C with one of the primary antibodies listed below: ALK (Cat. no. 51-390 Invitrogen); phospho-Tyrosine MultiMab™ (Cat. no. 8954, Cell Signaling Technology) (1:1000); phospho-Akt (Thr308) (Cat. no. 2965, Cell Signaling Technology) (1:1000); Akt1/2/3 (sc-8312, Santa Cruz Biotechnology) (1:1000); extracellular signal-regulated kinases 1 and 2 (ERK1/2) (cat no. 9102, Cell Signaling Technology); phospho-ERK1 (Thr202/Tyr204)/ERK2 (Thr185/Tyr187) (cat no. RA15002, Neuromics, Nothfield, MN, USA) (1:2000); actin (Cat. no. A2066, Sigma-Aldrich) (1:200 phospho-AMP-activated protein kinase (AMPK) (thr172) (cat. No. 2535, Cell Signaling Technology); and total AMP-activated protein kinase (AMPKα1/2) (sc-25792, Santa Cruz Biotechnology, Dallas, TX, USA) (1:1000). Then, the membranes were washed and exposed to a suitable horseradish peroxidase-conjugated secondary antibody (Santa Cruz Biotechnology, Dallas, TX, USA) by using Clarity Western ECL substrate (Bio-Rad Lab, Hercules, CA, USA). Immunoreactive bands were detected by the Luminescence Image analyser LAS 4000 (FujiFilm, Tokyo, Japan), and the optical densities were measured by using the NIH ImageJ software (US National Institutes of Health, Bethesda, MA, USA). The density of the matching total protein in the same sample was used to standardize the density of the phosphorylated protein bands. For the remaining immunoblots, the actin levels were used to standardize the densitometric results, as indicated. 

### 2.6. Immunoprecipitation

Immunoprecipitation was performed as previously described [31]. Hypothalamus tissues were sonicated with ice-cold RIPA buffer by adding 1% Triton X 100. After being spun at 10,000× *g* for 10 min at 4 °C, the supernatant (around 800 g of protein) was incubated with anti-phospho-Tyrosine antibody (1:200) (cat. No. 8954, Cell Signaling Technology) or pre-immune rabbit IgG (1:100) (Santa Cruz Biotechnology, Dallas, TX, USA). Then, all samples were incubated at 4 °C for 3 h while being continuously rotated with 40 μL of Pure Proteome Protein G magnetic beads (Millipore, Burlington, MA, USA). Next, ice-cold PBS plus 0.1% Tween 20 buffer was used to wash the beads five times. After the last wash, we added 2× sample buffer, and the samples were heated for 5 min. 

### 2.7. Statistical Analysis

*Animals*: Body weight, food intake, and RWA were reported as mean  ±  SEM and were analyzed by two-way ANOVA for repeated measures with two factors being groups as a between-subjects factor and time (days) as a within-subjects factor and a repeated factor followed by Bonferroni’s multiple comparisons test. *Plasma levels*: Data are shown as the mean ± SEM, and Student’s *t*-test was used for the statistical analysis. *Western Blotting*: Data are shown as the mean ± SEM. Results are shown as a percentage or fold stimulation of the control, which was a part of each separate experiment unless otherwise stated. The control group was set at 100, and a variance was calculated by expressing each control value as a percentage of the raw control group mean. The variance of this value was determined in the same manner as for the control group. Statistical analysis was performed by using a one-way analysis of variance (ANOVA) followed by Tukey’s test. Graph Pad 8 was used to apply the statistical analysis (San Diego, CA, USA). Differences that showed a *p* < 0.05 were considered significant. The statistical analysis and results are in accordance with the pharmacology experimental design and analysis guidelines [32].

## 3. Results

### 3.1. First ABA Induction 

#### 3.1.1. Body Weight, Food Intake, and RWA

During the 6 days of restricted feeding schedule, ABA and restricted animals significantly reduced their body weight as related to control and exercise rats (Figure 2A). Two-way ANOVA analysis reported a main significant group × time interaction effect [F (18, 144) = 76.43, *p* < 0.05]. However, on the last day of the restricted feeding schedule, ABA rats lost around 20% of their initial weight, while restricted rats showed only a modest but significant weight loss of 5%. In contrast, control and exercise rats, which were fed *ad libitum*, showed growth in body weight with no significant difference between the two groups. Due to the restricted feeding schedule, ABA and restricted rats ate less daily chow than ad libitum-fed animals and two-way ANOVA reveals a significant main effect of group × time interaction [F (15, 120) = 10.29, *p* < 0.05] (Figure 2B). The loss of weight in ABA animals was concomitant with a progressive intensification in RWA, which varies significantly from that of exercise rats. Two-way ANOVA reveals a significant main effect of group × time interaction [F (6, 72) = 2.66, *p* < 0.05] (Figure 2C). On day 6, the RWA of the ABA rats showed an increase of 89% compared to the baseline (BL). In contrast, exercise rats reported an increase of 42% from BL.

#### 3.1.2. Adiponectin Levels

Adiponectin is a hormone secreted by adipose tissue that regulates food intake and energy expenditure through its action in the hypothalamus [33]. It is well established that plasma levels of adiponectin are higher in AN than in control women, thus suggesting that hyperadiponectinemia could contribute to the pathogenesis of AN [34]. In agreement with human studies, our ABA rats showed higher adiponectin plasma levels than control animals (unpaired *t*-test: t(12) = 2.758, *p* < 0.05) (Figure 2D).

#### 3.1.3. ALK Receptor Signaling

To evaluate the impact of the anorexic phenotype on ALK receptor signaling in the hypothalamus, we first examined its expression by Western blot analysis. Analysis of the ALK receptor was performed with an anti-ALK receptor antibody, which detected the presence of an immunoreactive band of 220 KDa, corresponding to the molecular mass of the full-length isoform [35]. As shown in Figure 3A, ALK protein expression was significantly reduced (33%) in ABA rats [one-way ANOVA *p* < 0.05]. To gain insight into ALK receptor phosphorylation status in the ABA paradigm, we performed an IP assay. An immunoreactivity band with no discernible alterations in the phosphorylated tyrosine status was seen after the receptor immunoprecipitation, as shown in Figure 3B.

The ALK receptor belongs to the IGF receptor family, and it is known to trigger the phosphatidylinositol 3-kinase (PI3K) pathway and its downstream effector, serine/threonine kinase Akt [36]. To further characterize the ALK receptor signaling, we explored the ALK downstream PI3K/Akt pathway, measuring the levels of Akt phosphorylation at thr^308^. Figure 3C illustrates a significant decrease (~28%) in Akt phosphorylation at thr^308^ in ABA rats compared to control rats [one-way ANOVA *p* < 0.05]. These data suggest that the ABA model is linked to changes in the ALK’s levels and its downstream signaling cascade. 

Previously, Murugan’s group demonstrated that ALK mutation might affect the mitogen-activated protein (MAP) kinase pathway [37]. Moreover, in transfected PC12 cells, activation of ALK causes a rapid and persistent activation of MAP kinase [38]. The extracellular signal-regulated kinase (ERK) controls major cellular processes, such as differentiation, proliferation, and migration. Once phosphorylated, ERK may translocate to the nucleus and activate gene expression through specific transcription factors. As shown in Figure 3D, no discernible alterations in ERK1/2 phosphorylation were found in any of the analyzed experimental groups, [One-way ANOVA *p* = N.S.]. 

#### 3.1.4. AMPK Phosphorylation and Caspase Activation

It is known that AMPK is a serine/threonine protein kinase, a key cellular sensor to regulate the homeostasis of energy. When AMPK is activated and phosphorylated, cell energy decreases to ensure the normal functions of physiological cell activities [39]. AMPK is a mediator of adaptive responses in the hypothalamus, regulating feeding behavior with modifications of nutrients and hormones. As presented in Figure 4A, only the restricted rats demonstrated an enhanced level of AMPK phosphorylation, while exercise and ABA rats did not show a significant increase [one-way ANOVA *p* < 0.05]. This finding would fit well with the catabolic state that occurs during starvation, where the energy to build new cells would be scarce. Moreover, exercise and the ABA group share free access to a running wheel that could affect the production of substances that could offset the energy expenditure brought on by exercise.

It has been shown that in AN subjects, abdominal fluid, and pericardial effusion are reduced [40,41]. It is quite likely that these fluid movements in patients with malnutrition may trigger apoptosis in brain cells, such as glial or neuronal cells, because similar changes in oncotic pressure are also mirrored in the brain [41]. Moreover, it is known that the ALK receptor has a proapoptotic activity even in the absence of ligands [42]. By virtue of the alterations that have been observed in AN patients in relation to apoptosis, we investigated the executioner caspase-3 and poly-(ADP ribose) polymerase (PARP), a downstream target of activated caspases. Using the Western blotting method, caspase 3 was not revealed in its cleaved form (Figure 4B), while the native PARP form did display a faint band without a cleaved one. Furthermore, our data showed that the ABA paradigm failed to cause apoptosis and that the decrease of the ALK receptor was unrelated to the activation of the caspase pathway [one-way ANOVA *p* = N.S.]. 

### 3.2. Recovery Phase

#### Body Weight, Food Intake, RWA, and ALK Receptor Expression

In a different group of animals, alterations of ALK expression were assessed after a 7-day period of recovery from the ABA induction to evaluate whether any alterations persisted after weight recovery. During the recovery phase, ABA and restricted rats were allowed chow ad libitum. Indeed, they recovered the body weight loss to the basal value (100%) at day 2. However, on day 7 of the recovery phase, ABA and restricted rats still weighed significantly less than the exercise rats (Figure 5A). Two-way ANOVA showed a main significant group × time interaction effect [F (18, 144) = 15.5, *p* < 0.05]. Significant differences in body weight were also observed between control and exercise rats during this phase. During the recovery, food intake of ABA rats was significantly higher than the other experimental groups. Two-way ANOVA revealed a significant main effect of group × time interaction [F (18, 144) = 11.26, *p* < 0.05; Figure 5B]. Moreover, exercise rats consumed significantly more food compared with control animals. Although two-way ANOVA did not show a significant group x time interaction [F (6, 72) = 0.9148, *p* = N.S.], the running activity of ABA rats was significantly lower in comparison to the exercise rats during the first 3 days of the recovery phase (Figure 5C). During the recovery from body weight loss, ALK levels in ABA rats returned to the control baseline values, as shown in Figure 5D.

### 3.3. Second ABA Induction

#### Body Weight, Food Intake, RWA, and ALK Receptor Expression

In a third group of animals, after the refeeding period, we again exposed the ABA animals to a second new ABA protocol to evaluate ALK levels in a condition that could mimic a relapse of the disease [27]. During the second ABA induction, we found that ABA and restricted animals demonstrated a percentage of weight loss not dissimilar to that of the first ABA induction (Figure 6A). In contrast, compared to the first induction, the body weight gain of exercise rats was significantly larger than that of the control group. Two-way ANOVA analysis reported a significant main effect of group × time interaction [F (18, 144) = 105.8, *p* < 0.05]. Moreover, exercise rats consumed more chow than control rats, while the chow intake of ABA and restricted animals was lower in comparison to the other two groups of rats. Two-way ANOVA analysis showed a significant main effect of group × time interaction [F (15, 120) = 35.08, *p* < 0.05; Figure 6B]. RWA of ABA rats was significantly higher in comparison to the exercise group, and two-way ANOVA showed a significant main effect of group x time interaction [F (6, 72) = 4.417, *p* < 0.05; Figure 6C]. In the second induction, the reduction in ALK was marked but less than in the first (Figure 6D; *p* < 0.05). A possible explanation could be an adaptation and a less important impact given by a second cycle of ABA.

## 4. Discussion

Our research examined the hypothalamic ALK receptor and cell signaling in relation to the rats’ body weight changes to understand the mechanisms behind the alterations brought about by weight loss in AN. AN is a disease closely related to psychiatric phenotypes and is regulated by the energy balance circuitry in the hypothalamus, which contains integrative systems that support life, such as food intake, energy expenditure, and reproduction. The behavioral outcomes of the ABA model’s induction and recovery phases are similar to data reported in our earlier studies and corroborate the reliability of the ABA paradigm [27]. Indeed, it is possible to observe both physical hyperactivity and concomitant weight loss in young female adolescent rats when subjected to a planned dietary restriction combined with free access to a running wheel. It is well established that adiponectin plasma levels are significantly greater in AN than in control women, thus, the hyperadiponectinemia could contribute to the pathogenesis of AN [34]. In agreement with human studies, our ABA rats showed higher adiponectin plasma levels. Notably, adiponectin levels were increased in ALK-deficient mice [20]. Adiponectin is a hormone secreted by adipose tissue that possesses a pivotal role in energy homeostasis by regulating food intake and energy expenditure through its action in the hypothalamus [33]. In contrast, adiponectin levels are reduced in obese humans [43,44].

Our study clearly demonstrated a reduction of the ALK receptor level in the ABA group, which appears to confirm the hypothesis of its involvement in thinness. As already mentioned, genetic deletion of ALK caused thinness in mice with clear resistance to diet-induced obesity [20]. Moreover, these animals showed elevated daily energy expenditure, reduced adiposity, and improved glucose tolerance. Orthofer and collaborators also reported that ALK expression in hypothalamic neurons controls energy expenditure via sympathetic control of adipose tissue lipolysis [20]. So far, several potential human ALK ligands able to activate the receptor, such as augmentor-α, augmentor-β, pleiotrophin, and heparin, have been discovered [13,14,15,16]. Knockout mice of augmentor-α exhibit a similar thinness phenotype and resistance to diet-induced obesity of ALK knockout mice [21]. Moreover, in the augmentor-α knockout mice, the leanness phenotype was combined with increased physical activity. Also, augmentor-α-deficient mice showed reduced ALK activation in the paraventricular nucleus (PVN) [21].

Our data also showed that the combination of diet restriction with physical exercise was conducive to marked alterations not only in ALK receptor expression but also in the levels of downstream receptor signaling. In general, ALK is a member of the insulin receptor superfamily, and activates different signaling pathways, such as the PI3K-AKT, JAK-STAT, and MAPK pathways. A large body of genetic and pharmacological evidence indicates that the PI3K/Akt signaling pathway is critical for cell growth and survival [45]. The downregulation of Akt signaling in the ABA group was likely due to a decreased expression of ALK receptor and data that go in the same direction as those obtained with ALK. 

ALK mutations may affect the MAP kinase pathway; in fact, according to Murugan’s [37] and Souttou’s studies, it has been shown that a chimeric protein that mimics the ligand binding of ALK activates the MAP kinase pathway [38]. The MAP kinase family includes the ERK1/2, which plays a major role in neural plasticity and cell survival. Our findings did not reveal any changes in ERK1/2 phosphorylation, proving that this pathway is not regulated by ALK signaling in our system.

Considering these results, we concentrated the study on the AMPK protein. This kinase is a key enzyme in the regulation of energy metabolism, and it is an evolutionarily conserved enzyme that senses the energy status of the cell and regulates fuel availability [39]. The role of AMPK in the regulation of body weight and energy homeostasis is not limited to its actions in the peripheral tissues and is widely expressed in the brain, including the hypothalamus, where it works by influencing energy intake and availability [46]. Our data demonstrate a considerable rise in AMPK phosphorylation only in rats given a limited food plan (restricted). The ABA group, in contrast to the restricted one, shows no discernible increase in AMPK levels. Previous studies have shown discrepancies in AMPK activity in different ABA models [39,47,48,49].

One explanation for the divergence in the literature may rely on the techniques used to cause AN, the length of the treatment, and the variety in the animal species used, such as mouse or rat. The increase of the phosphorylation of AMPK in the restricted group demonstrates how reduced caloric intake activates a signaling pathway in the hypothalamus, to compensate for the energy unbalance, and makes available the energy missing from the reduced food regimen. In addition, the ABA group already has an activity aimed at “thinness” given by the reduction of the ALK protein and the increase in AMPK activity could collide with the thinness mechanism triggered by ALK. Of course, multiple studies are still needed to define and clarify the role of AMPK in AN.

Moreover, ALK is also known to be a dependence receptor, which may itself activate the apoptosis machinery. It is also known that, once activated, caspase 3 may be the cause of ALK degradation [42]. However, it must also be considered that the lower levels of ALK may also be mediated directly by the extreme ABA condition, which might activate the apoptosis. All these considerations prompted us to investigate the activation of the executioner caspases 3 and the cleavage of PARP. Our data demonstrated that the ALK receptor does not undergo caspase-3- or PARP-supported cleavage because it is not activated in ABA as in all other experimental groups. Although the ABA model represents one of the most used animal models for the study of AN, it is important to underline that not all the traits that characterize human pathology can be reproduced [50]. For example, it cannot replicate the psychological components, such as the intense fear of gaining weight and an altered relationship with the shape of one’s body, which leads to a state of extreme thinness. Anyway, the results of this study demonstrated, for the first time, that the ALK receptor may be able to regulate anorexia changes. Its role in AN remains to be further elucidated, but our findings are probably a reflection of a relationship between the decrease in body weight, the reduction of the ALK receptor, and, in consequence, an increase of status of thinness, which could be the missing link to explain the state of continued thinness even after an anorexic subject starts feeding again.

## Figures and Tables

**Figure 1 nutrients-15-02205-f001:**
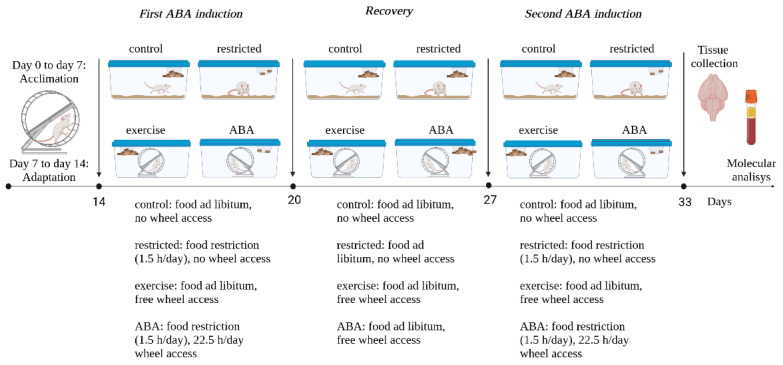
Schematic representation of the experimental timeline.

**Figure 2 nutrients-15-02205-f002:**
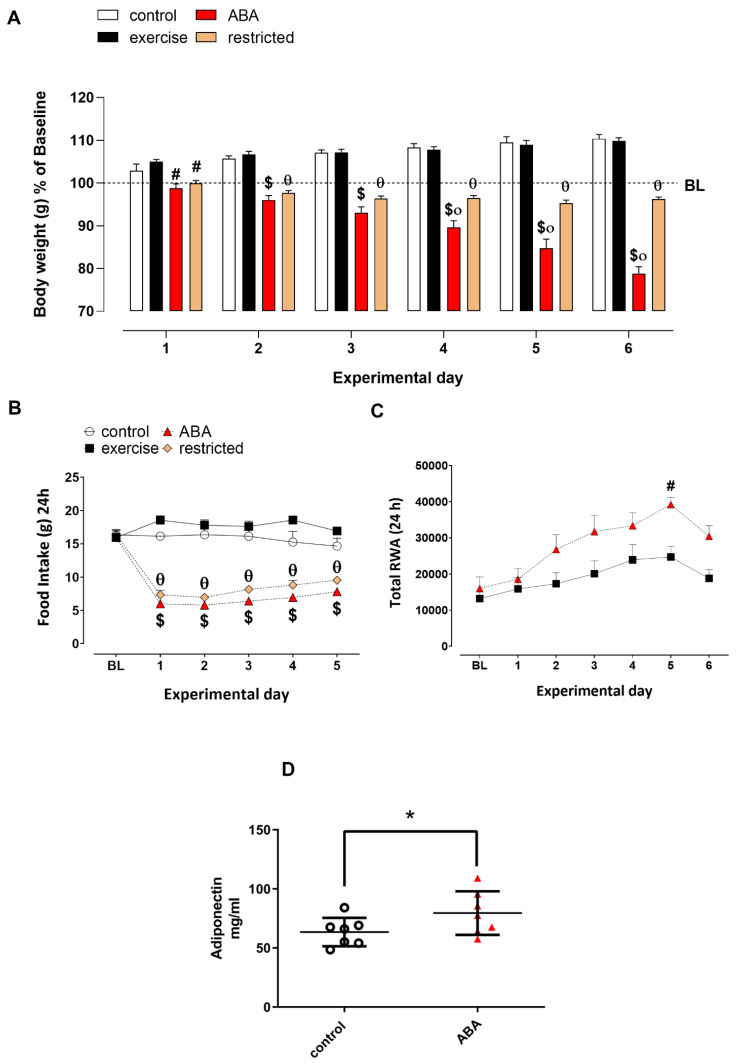
Body weight (**A**), food intake (**B**), and RWA (**C**) in control, exercise, restricted, and ABA groups during the 6 days of the first ABA induction. Results are presented as the mean ± SEM (*n* = 7 rats per group). Statistical analysis were performed by two-way ANOVA for repeated measures with two factors being groups as a between-subjects factor and time (days) as within-subjects factor and a repeated factor followed by Bonferroni post hoc test (Body weight: # *p* < 0.05 vs. exercise rats, $ *p* < 0.05 and θ *p* < 0.05 vs. control and exercise groups, ο *p* < 0.05 vs. restricted; food intake: $ *p* < 0.05 vs. control and exercise, θ *p* < 0.05 vs. control and exercise, ο *p* < 0.05 vs. restricted; RWA: # *p* < 0.05 vs. exercise). (**D**) Adiponectin levels were significantly increased in the blood of the ABA groups with respect to the control rats * *p* < 0.05, analysis was performed by the Student’s test.

**Figure 3 nutrients-15-02205-f003:**
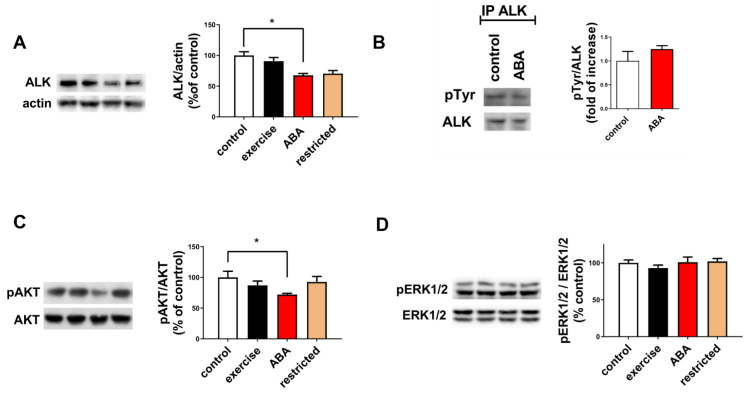
Effect of ABA on ALK, pAKT and pERK1/2. (**A**) Protein level of the hypothalamus (*n* = 6 rats for ALK), * *p* < 0.05 versus control, by one-way ANOVA followed by Tukey’s test. (**B**) Protein level of pAKT thr308 in the hypothalamus (*n* = 6 pAKT), * *p* < 0.05 versus control, by one-way ANOVA followed by Tukey’s test. (**C**) ALK was immunoprecipitated (IP) with a specific antibody, and the level of tyrosine phosphorylation was determined by Western blot (WB) using an antiphospho-tyrosine antibody (pTyr) (*n* = 3 rats). Values are the mean  ±  SEM of three experiments. (**D**) Protein level of p ERK1/2 in the hypothalamus (*n* = 5 rats).

**Figure 4 nutrients-15-02205-f004:**
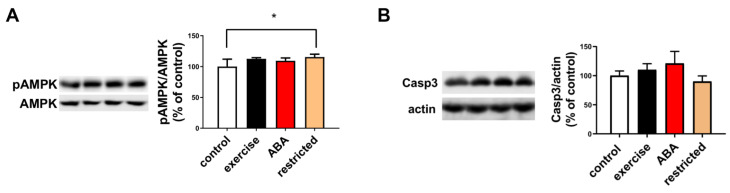
Effect of ABA on pAMPK and caspase3. (**A**,**B**) Protein level of pAMPK (*n* = 5 rats), and caspase3 in the hypothalamus (*n* = 7 rats), * *p* < 0.05 versus control, by ANOVA followed by Tukey’s test.

**Figure 5 nutrients-15-02205-f005:**
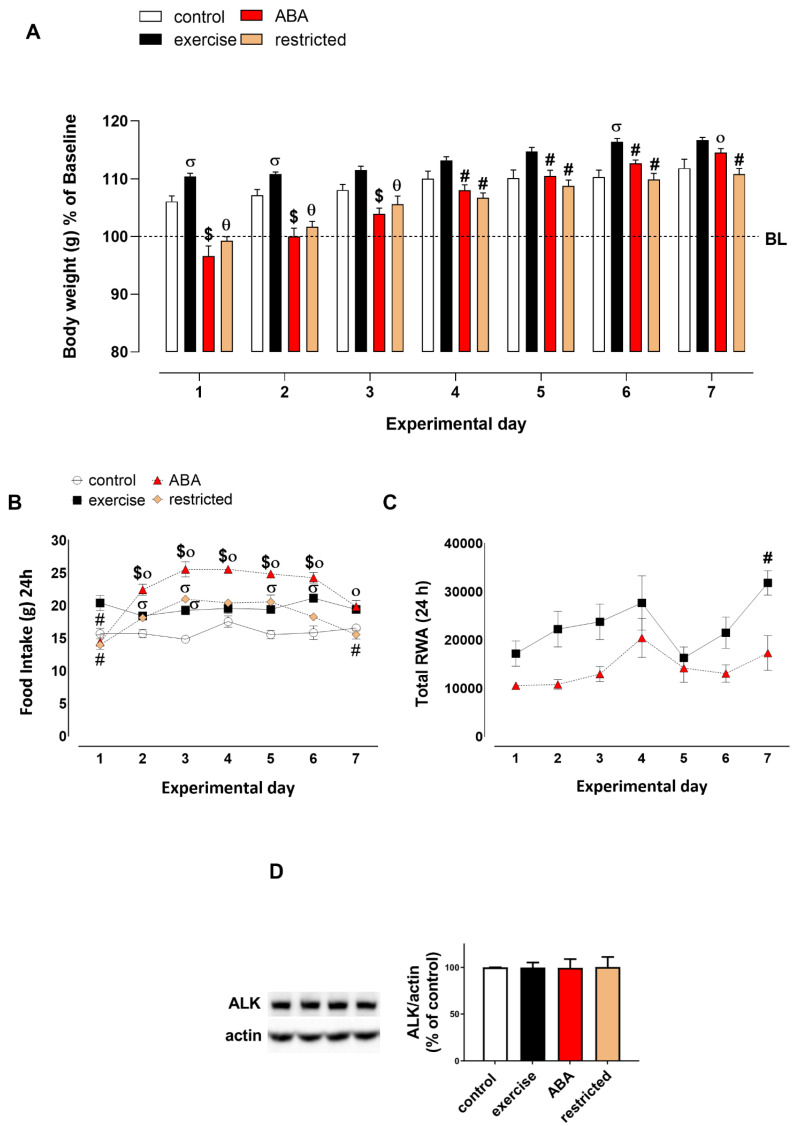
Body weight, food intake, and RWA, in control, exercise, restricted, and ABA groups during the recovery (**A**–**C**). Results are presented as the mean ± SEM (*n* = 7 rats per group). Data were analyzed using two-way ANOVA followed by Bonferroni post hoc test for repeated measures (Body weight: # *p* < 0.05 vs. exercise rats, $ *p* < 0.05 and θ *p* < 0.05 vs. control and exercise groups, ο *p* < 0.05 vs. restricted, σ *p* < 0.05 vs. control; food intake: σ *p* < 0.05 vs. control, $ *p* < 0.05 vs. control and exercise, θ *p* < 0.05 vs. control and exercise, ο *p* < 0.05 vs. restricted; RWA: # *p* < 0.05 vs. exercise). (**D**) Protein level of ALK in the hypothalamus by ANOVA, followed by Tukey’s test (*n* = 6 rats).

**Figure 6 nutrients-15-02205-f006:**
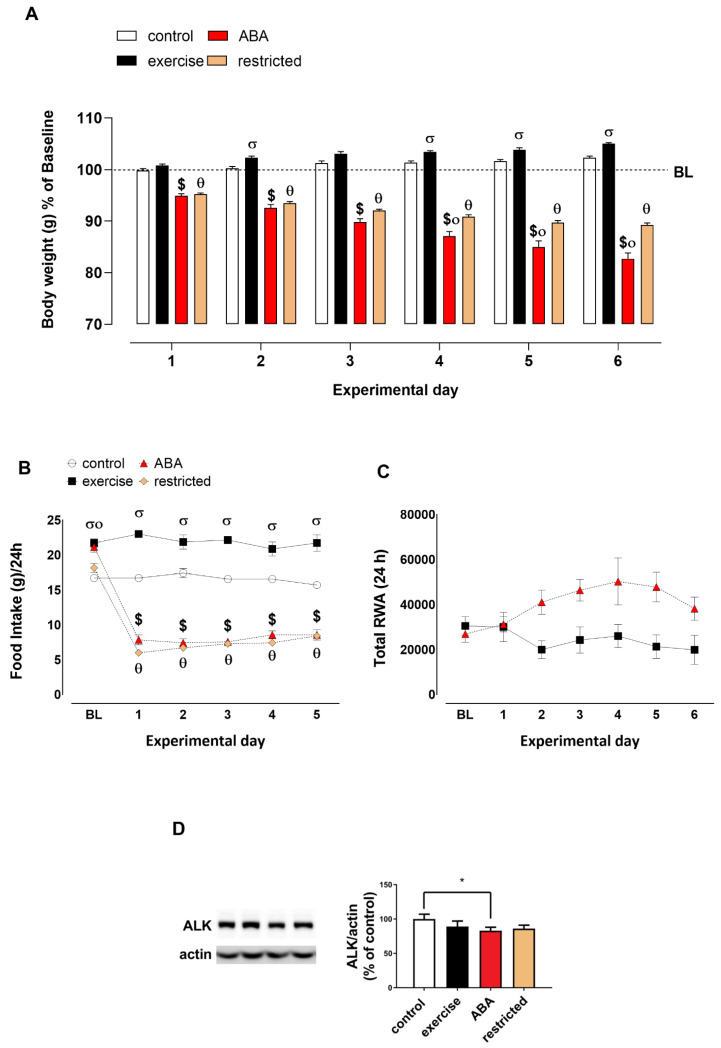
Body weight, food intake, and RWA, in control, exercise, restricted, and ABA groups during the second ABA induction (**A**–**C**). Results are presented as the mean ± SEM (*n* = 7 rats per group). Data were analyzed using two-way ANOVA followed by Bonferroni post hoc test for repeated measures (body weight: # *p* < 0.05 vs. exercise rats, $ *p* < 0.05 and θ *p* < 0.05 vs. control and exercise groups, ο *p* < 0.05 vs. restricted, σ *p* < 0.05 vs. control; food intake: σ *p* < 0.05 vs. control, $ *p* < 0.05 vs. control and exercise, θ *p* < 0.05 vs. control and exercise, ο *p* < 0.05 vs. restricted). (**D**) Protein level of ALK in the hypothalamus (*n* = 6 rats), * *p* < 0.05 versus control, by ANOVA followed by Tukey’s test (*n* = 6 rats).

## Data Availability

Not applicable.

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
