# Peer review of "Anaplastic Lymphoma Kinase Receptor: Possible Involvement in Anorexia Nervosa"

_nutrients, 2023, doi:10.3390/nu15092205_

Round 1
Reviewer 1 Report
This is an interesting and well-written manuscript reporting a study where anaplastic lymphoma kinase (ALK) receptor expression was measured in rats of the activity-based anorexia model. A down-regulation of the ALK receptor expression was found. This finding was reversible with weight recovery.
I have only a few minor suggestions for improvement:
- The total number of rats and the number of rats in each group should also be mentioned in the abstract. Otherwise, the reader has no idea about the sample size of this study.
- Sometimes, the authors write "n=6", sometimes they write "n=6 rats". This should be consistent "n=6 rats" throughout the manuscript. The "6" is only an example for a number.
- "Control" is sometimes capitalized, sometimes not. "control" or "controls" should not be capitalized throughout the manuscript. The same applies to "Exercise" or "Exercise group". I don't understand why this is sometimes capitalized.
- In Figure 6 it says: "(D) Protein level 349 of ALK in the hypothalamus (n= rats)". Please insert the number of rats.
- In order to avoid further similar formal mistakes, the authors should check the whole manuscript regarding the format.
- There is no limitation section in the discussion. However, limitations should be mentioned, for example, the problems of the activity-based anorexia model and the small number of rats should be discussed as a shortcoming.
Author Response
Referee 1
We thank the Reviewer for the positive comments.
- The total number of rats and the number of rats in each group should also be mentioned in the abstract. Otherwise, the reader has no idea about the sample size of this study.
Response 1: We are sorry, but since the experimental groups are not mentioned in the abstract, we do not deem it necessary to insert the number of rats used for each of them.
- Sometimes, the authors write "n=6", sometimes they write "n=6 rats". This should be consistent "n=6 rats" throughout the manuscript. The "6" is only an example for a number.
Response 2: We have revised and corrected the text as suggested by the referee.
- "Control" is sometimes capitalized, sometimes not. "control" or "controls" should not be capitalized throughout the manuscript. The same applies to "Exercise" or "Exercise group". I don't understand why this is sometimes capitalized.
Response 3: We thank the reviewer for highlighting this confounding point and we have modified the text accordingly. We have also corrected all the figures according to the manuscript text.
- In Figure 6 it says: "(D) Protein level 349 of ALK in the hypothalamus (n= rats)". Please insert the number of rats.
Response 4: We have modified as suggested by the referee.
- In order to avoid further similar formal mistakes, the authors should check the whole manuscript regarding the format.
Response 5: We checked the whole manuscript as suggested by the referee.
- There is no limitation section in the discussion. However, limitations should be mentioned, for example, the problems of the activity-based anorexia model and the small number of rats should be discussed as a shortcoming.
Response 6: We thank the reviewer for highlighting this point and we have now addressed this issue in the conclusions of the discussion as follow:
“Although the ABA model represents one of the most used animal models for the study of AN, it is important to underline that not all the traits that characterize human pathology can be reproduced [50]. For example, it cannot be able to replicate the psychological components such as the intense fear of gaining weight and an altered relationship with the shape of one's body, which leads to a state of extreme thinness.”

Reviewer 2 Report
Line 34: "such"as genetic... . Line 38: Should be "Adolescents". Line 56 "other" rather than other. Line 90: possible "parameters" to reduce..... .Line 109: "an" established.....
Figure 1: A good figure , yet difficult to read. I am hoping that it will be clearer when the manuscript is published.
Line 133: Should be "1.5", rather than 1,5. Line 141: "was then" centrifuged ... . Line 143: "using a commercially available ELISA kit", or, "commercial"
Line 191: "we" added .... . Line 193: "Animals" Line198: data are "shown"; Results are "shown" ..... . Line 205: were "considered" significant, rather than stated. Line 217: "ate" rather than eaten.
Figure 2 is well done and clear, but Line: 232 : ABA groups "with" respect" to . Line 235: "regulates" food..... Line 236: in "the " hypothalamus Line 248 : "we" performed...
Figures 3 and 4 are also clear, however, they should be labeled on the Figures, for example (A) , (B), etc.
Lines 258 : belongs to "the"; Line 259 "(PI3K) "pathway" Line 307: "weighed" . Figure 5 is clear, and easy to follow.
Line 331: "demonstrated". Line 333 "larger". Line 337 : "groups".
Figure 6, well displayed, I have no problems with it. Line 369: receptor tone ? clarify. Line 424 : "The results of this study demonstrated, for the first time, ...... .
This manuscript addresses an important concept in addressing the relationship between Anorexia nervosa, and, anaplastic lymphoma kinase receptor. This may have importance in a better understanding of anorexia.
As such, it may lead to better intervention and treatment of Anorexia Nervosa.
I would like to see this manuscript published after revision.
A well written manuscript, after the revisions have been made. The figures are clear for the most part, with minor changes (which can be remedied) as suggested in the revision of the manuscript.
Author Response
Referee 2
We appreciate the Reviewer's helpful comments.
- Line 34: "such"as genetic... . Line 38: Should be "Adolescents". Line 56 "other" rather than other. Line 90: possible "parameters" to reduce..... .Line 109: "an" established.....
Response 1: We have modified as suggested by the referee.
- Figure 1: A good figure, yet difficult to read. I am hoping that it will be clearer when the manuscript is published.
Response 2: As suggested, we have edited and changed the figure 1 hoping to make it clearer.
- Line 133: Should be "1.5", rather than 1,5. Line 141: "was then" centrifuged ... . Line 143: "using a commercially available ELISA kit", or, "commercial"
Response 3: We have modified as suggested by the referee
- Line 191: "we" added .... . Line 193: "Animals" Line198: data are "shown"; Results are "shown" ..... . Line 205: were "considered" significant, rather than stated. Line 217: "ate" rather than eaten.
Response 4: We have modified as suggested by the referee.
- Figure 2 is well done and clear, but Line: 232 : ABA groups "with" respect" to . Line 235: "regulates" food..... Line 236: in "the " hypothalamus Line 248 : "we" performed...
Response 5: We have modified as suggested by the referee
- Figures 3 and 4 are also clear, however, they should be labeled on the Figures, for example (A) , (B), etc.
Response 6: We are very sorry, we checked, and the figures appear labeled.
- Lines 258 : belongs to "the"; Line 259 "(PI3K) "pathway" Line 307: "weighed" . Figure 5 is clear, and easy to follow.
Response 7: We have modified as suggested by the referee
- Line 331: "demonstrated". Line 333 "larger". Line 337 : "groups".
Response 8: We have modified as suggested by the referee.
- Figure 6, well displayed, I have no problems with it. Line 369: receptor tone ? clarify. Line 424 : "The results of this study demonstrated, for the first time, ...... .
Response 9: We have modified and clarified as suggested by the referee
